# FOCUS: Forcing In-Context Object Localization through Visual Support Constraints and Policy Optimization

**Mohammed Asad Karim** [* 1]   **Vinay Kumar Verma** [* 1]

## Abstract

In-context localization (ICL) seeks to localize a target object specified by a small set of support examples in a query image, operating on the fly without training or parameter updates. Despite rapid advances in vision–language models (VLMs), achieving category-agnostic and visually grounded ICL remains an open problem, even though it is essential for applications such as image editing, personalized visual search, and retrieval. Existing methods are fragile and rely on explicit category supervision, which not only limits applicability in realistic settings with unnamed or instance-specific objects but also introduces category bias that steers predictions toward semantic priors rather than visual evidence. We introduce a two-stage training framework that explicitly optimizes in-context attention between support bounding boxes and query images without category supervision. We further refine localization via reinforcement learning using Group Relative Policy Optimization (GRPO) to directly minimize localization error. This formulation enforces visual correspondence over semantic priors, yielding robust instance-level localization. Empirically, a 7B-parameter model trained with our objectives outperforms models up to 72B parameters, demonstrating that context-aware localization objectives can surpass scaling alone. Comprehensive ablations validate the contribution of each component.

## 1. Introduction

Vision–language models (VLMs) (Wang et al., 2024; Li et al., 2024; Dai et al., 2023a; Li et al., 2023) have achieved remarkable success across a broad spectrum of vision and language tasks, yet they continue to struggle with *in-context object localization*. In-Context Object Localization (ICOL) focuses on localizing a user-specified object in a query image by relying solely on a small set of visual support examples available at inference time. In contrast to traditional object detection or grounding approaches that depend on fixed category vocabularies and extensive supervised training, ICOL enables models to infer the target concept on the fly, without parameter updates, by reasoning over visual correspondences between support and query images. This capability is critical for practical applications such as customized image editing, personalized visual search, and interactive object tracking, where the object of interest is user-defined, instance-specific, and often difficult or impossible to describe textually in advance. Achieving reliable in-context localization is therefore a key step toward flexible, user-driven visual understanding systems.

Despite the success of in-context learning in large language models (LLMs) (Alayrac et al., 2022; Brown et al., 2020b; OpenAI, 2023; Raffel et al., 2020), transferring this paradigm to vision–language models (VLMs) for object localization remains challenging. Recent work on in-context object localization shows that VLMs (Singh et al., 2022) can use support examples with bounding box annotations to localize novel objects at inference time; however, their predictions are still strongly shaped by category-level priors rather than instance-specific visual reasoning (Doveh et al., 2025). To address this, Doveh et al. introduce pseudo-labels during training to reduce reliance on true category names and encourage visual grounding. In practice, inference continues to rely on original category labels, creating a train–test mismatch that reintroduces semantic priors at deployment. As a result, category bias is only partially mitigated, and localization decisions can still be driven by semantic cues instead of the visual evidence provided by support examples. Moreover, empirical findings suggest that even with pseudo-label training, models frequently underutilize fine-grained spatial cues such as bounding box geometry and relative position defaulting to coarse visual similarity or residual category correlations. These observations reveal a central gap: existing approaches fail to eliminate category-mediated reasoning and do not enforce strict reliance on visual support constraints for instance-level localization.

[1]Amazon, Seattle, USA; [*]Equal contribution. Correspondence to: Mohammed Asad Karim <asadkarim0938@gmail.com>.

*Proceedings of the 43rd International Conference on Machine Learning*, Seoul, South Korea. PMLR 306, 2026. Copyright 2026 by the author(s).

To address these limitations, we propose a category-agnostic, attention-based formulation of in-context object localization that removes category names entirely from both support and query prompts. Instead of relying on true labels or pseudo-labels, our method optimizes in-context attention using only visual support examples, ensuring that localization decisions are driven by visual correspondence and spatial reasoning. By removing textual identifiers, the model is no longer influenced by semantic category priors and must infer the target object from visual appearance and geometry. This attention-based design promotes direct use of bounding box information, encouraging consistent focus on object shape, relative position, and surrounding visual cues across support and query images. To further improve query alignment, we refine bounding box prediction using GRPO-based reward optimization, which directly minimizes alignment error. As a result, the model generalizes to arbitrary object instances, including unseen categories and visually defined concepts without stable names. Overall, our approach reframes in-context localization as a visual reasoning task, enabling robust and generalizable instance-level grounding. We summarize our key contributions as follows:

- We propose a category-independent, pure visual context–based in-context localization framework that overcomes category-induced bias in VLMs.

- We introduce an attention map optimization that encourages the model to focus on the most relevant regions in both support and query images for robust localization. In addition, a GRPO-based reward objective is used to reduce object bounding-box alignment error.

- Through extensive experiments, we demonstrate that the proposed model (*FOCUS*) enables effective in-context localization without relying on category labels or prior semantic information.

## 2. Related Work

**In-context learning in language models.** In-context learning (ICL) (Doveh et al., 2025) enables large language models (LLMs) to perform new tasks by conditioning on a small set of demonstrations at inference time, without parameter updates. Early work showed that few-shot performance scales with model and context size in autoregressive models (Brown et al., 2020a). Subsequent studies interpret ICL as implicit Bayesian inference (Xie et al., 2022) or gradient-descent-like computation emerging from transformer attention dynamics (Von Oswald et al., 2023). Empirically, ICL is sensitive to prompt composition (Min et al., 2022; Lu et al., 2022), motivating calibration (Zhao et al., 2021) and retrieval-augmented methods (Rubin et al., 2022). Instruction-tuned models further improve zero-shot generalization (Wei et al., 2022).

**In-context learning in vision–language models.** Compared to language models, in-context learning for vision–language models (VLMs) remains relatively underexplored (Alayrac et al., 2022; Zhang et al., 2024; OpenAI, 2023). Recent multimodal models demonstrate in-context capabilities for tasks such as visual question answering, reasoning, and visual grounding when examples are provided in the prompt (Alayrac et al., 2022; Brown et al., 2020b; Dai et al., 2023b). However, these demonstrations primarily focus on semantic understanding and rely on natural language to specify the task and the target object (Liu et al., 2024; Liao et al., 2025). As a result, in-context behavior in VLMs has largely been studied through language-conditioned interactions rather than through purely visual conditioning (Radford et al., 2021; Huang et al., 2023). A recent work IPLoc (Doveh et al., 2025) explores the ICL with the help of pseudo-label and visual grounding, however pseudo label reduce the generalization ability since for the novel category model have not learned the pseudo label.

**Semantic grounding and localization in VLMs.** Most vision–language models (VLMs) for grounding and localization rely on explicit semantic supervision, such as object names, attributes, or referring expressions. Early work formulates grounding as localizing regions described by natural language queries (Kazemzadeh et al., 2014; Yu et al., 2016), later extended to end-to-end multimodal transformers that condition detection on text (Kamath et al., 2021). Recent large-scale pretraining unifies open-vocabulary detection and phrase grounding by aligning visual regions with linguistic descriptions (Li et al., 2021; Zhang et al., 2022), while CLIP-based methods adapt contrastively trained models for grounding tasks (Xiao et al., 2023). Despite strong performance, these approaches assume targets can be specified unambiguously through language. This assumption limits applicability in settings with unnamed objects, visually similar instances, or domain-specific entities without canonical labels, motivating methods that relax or adapt text-conditioned grounding architectures (Shi et al., 2023).

**Few-shot and instance-level visual conditioning.** Few-shot object localization (Chen et al., 2022) has been studied via episodic and meta-learning across detection, segmentation, and tracking tasks (Bertinetto et al., 2016; Shaban et al., 2017; Yan et al., 2019). Recent work extends these paradigms to foundation and vision–language models using stronger visual and multimodal features (Madan et al., 2024; Han & Lim, 2024). However, most approaches rely on supervised adaptation or architectural updates rather than inference-time conditioning, and thus do not exhibit true in-context behavior (Xu et al., 2023; Liu et al., 2023).

**Pure visual in-context localization.** We introduce a pure visual in-context localization setting in which a single frozen

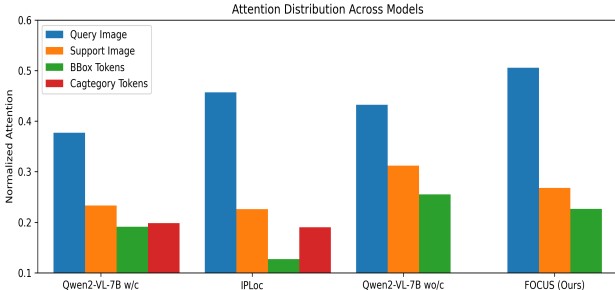

*Figure 1.* The figure illustrates in-context localization across different models by visualizing the support, query, predicted bounding boxes, and the corresponding attention maps. The support image provides a bounding box specifying the target object. Attention heatmaps highlight regions the model relies on for prediction, while red boxes indicate the final localized output.

*Figure 2.* Comparison of attention from answer tokens to input tokens. Our model places greater attention on query image tokens compared to the SFT baseline, indicating stronger visual grounding during localization. Here w/c and wo/c shows the model with and without category information respectively.

model localizes objects using only support images and bounding box annotations, without any semantic labels or textual cues. To our knowledge, this is the first systematic study of in-context localization driven solely by visual evidence. By eliminating linguistic priors, this formulation enables open-set generalization to novel object instances and enforces robust instance-level reasoning through visual correspondence alone.

## 3. Why LVMs Require Explicit Supervision?

We conduct an extensive empirical study to analyze failure cases of in-context localization models and design targeted objectives to address the identified limitations. Our key findings are summarized as follows:

**Prior Bias/ Category Name Biasness** We observe that language bias, particularly the use of category names, plays a decisive role in predicting the query bounding box. As illustrated in Fig. 1, IPLoc fails to localize the correct instance and instead attends to an incorrect bowl when multiple objects from the same category appear in the scene. This failure mode exposes a fundamental limitation of category-driven localization. When several instances share the same semantic label, category supervision biases the model toward visually salient or prototypical instances rather than the specific instance defined by the in-context example. Fig. 2 further supports this finding by examining attention from the answer token to the input tokens. We observe that category tokens receive comparable or even higher attention than

the relevant visual context, indicating reliance on semantic shortcuts instead of grounded visual reasoning.

**Attention Distribution to the Visual Context** We further analyze attention from the answer token to the input tokens, as shown in Fig. 2, which reflects each token's contribution during prediction. IPLoc and the vanilla w/c baseline assign limited attention to query and BBOX tokens while allocating relatively higher attention to category tokens, explaining their observed failure cases. The reduced attention to support and BBOX tokens indicates weak visual grounding. We then examine whether removing category names from the vanilla model (vanilla wo/c) resolves this issue. Although Fig. 2 suggests improved attention allocation across query, support, and BBOX tokens, this improvement is superficial. A closer inspection of Fig. 1 for Qwen2-VL-7B reveals that, without category information, the model exhibits high aggregated attention over the query image; however, this attention is diffusely distributed and fails to concentrate on regions corresponding to the support examples. Consequently, the predicted bounding boxes remain weakly grounded in the visual context.

Motivated by this analysis, we introduce an attention-based loss and a GRPO-based reward to correct these failure modes, explicitly enforcing support-aware visual attention and improving bounding box alignment for reliable in-context localization.

## 4. Preliminary

### 4.1. Problem Definition and Notation

This work study *in-context localization*, where a model localizes a target object in a query image by conditioning on a set of visual demonstrations (support images) provided within the same prompt. Let a sequence of images be:

$$\mathcal{I} = (I_1, I_2, \ldots, I_T), \tag{1}$$

where $\forall I_t \in \mathbb{R}^{H \times W \times 3}$ and $(I_1, I_2, \ldots, I_{T-1})$ are the support images that contain the target object annotated by the bounding box. For the support images, the bounding boxes are given as:

$$\mathcal{B}_{1:T-1} = \{b_1, b_2, \ldots, b_{T-1}\}, \tag{2}$$

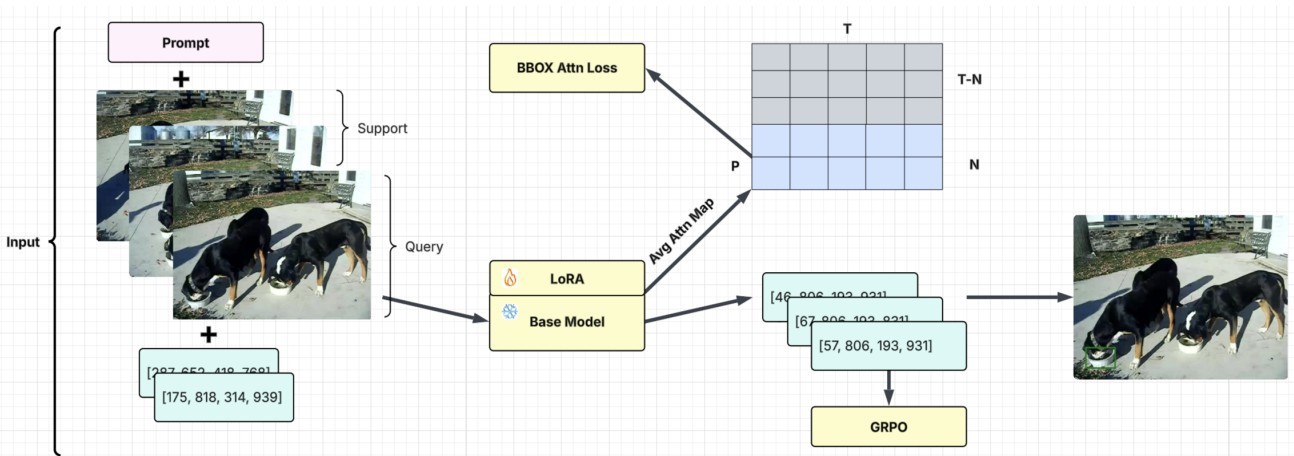

*Figure 3.* The block diagram of the proposed approach model (FOCUS): The model accepts the support set with the BBOX and predicts the final BBOX over the query image. Attention loss is applied to the attention map from the query to the input token, and GRPO helps generate a precise BBOX.

where each bounding box ($b_t$) is parameterized as $b_t = \left[x_{\min}^{(t)}, y_{\min}^{(t)}, x_{\max}^{(t)}, y_{\max}^{(t)}\right]$. Given this context, the objective is to predict the bounding box $b_T$ corresponding to the same object instance in the final query image $I_T$.

Let us assume that we have a multimodal autoregressive model $\mathbf{f}_\theta$, where $\theta$ is the model parameters. The model accepts the support image data and the bounding boxes along with the query sample as input and predicts the query object output as BBOX, which is defined as:

$$\hat{b}_T = \mathbf{f}_\theta(I_T/(I_1, I_2, \ldots, I_{T-1}), \mathcal{B}_{1:T-1}) \qquad (3)$$

where $\hat{b}_T$ is the predicted bounding box from the query image.

The model jointly encodes visual tokens extracted from images and textual tokens from the instruction prompt, enabling cross-modal reasoning over spatial relationships. Localization is performed by matching regions in the query image with the target object defined through bounding box demonstrations in the context. This formulation does not rely on temporal continuity, motion cues, or persistent object states. Each prediction is made independently based on in-context examples, framing localization as a demonstration-conditioned visual reasoning task.

## 5. Methodology

As outlined in Section 4, each training instance comprises multiple *support* images annotated with BBOXes, followed by a *query* image for which the model generates a BBOX prediction in text form. The proposed training framework consists of two stages. In the first stage, we optimize a category-agnostic, attention-based BBOX grounding objective using only visual context. This objective explicitly encourages attention concentration on relevant visual regions,

while suppressing attention to irrelevant areas and category-induced biases. In the second stage, we refine BBOX alignment through reinforcement learning using Group Relative Policy Optimization (GRPO), guided by an IoU-based reward. The following section presents a detailed description of the proposed approach.

### 5.1. Prompt Specification

All experiments use a fixed natural language prompt that defines the in-context localization task and specifies the expected output format. The prompt is prepended once to each input sequence, followed by interleaved images and their corresponding BBOX annotations. The exact prompt text used in all experiments is provided below:

**Prompt:** `Locate the same object across the sequence of frames shown below. Your goal is to identify the target object consistently using the visual context provided.`

`For the first` $T-1$ `frames, the bounding box of the object is already provided. Use this information and the visual context to predict where the same object appears in the final frame.`

`Output the predicted bounding box for the last frame in the following format:`

$$\langle\text{answer}\rangle[x_{\min}, y_{\min}, x_{\max}, y_{\max}]\langle/\text{answer}\rangle$$

Following the task description, each image in the sequence is paired with a corresponding BBOX annotation. For frames 1 through $T-1$, the input explicitly provides the bounding box of the target object associated with each image. For the final frame $T$, no bounding box is given, and the model is

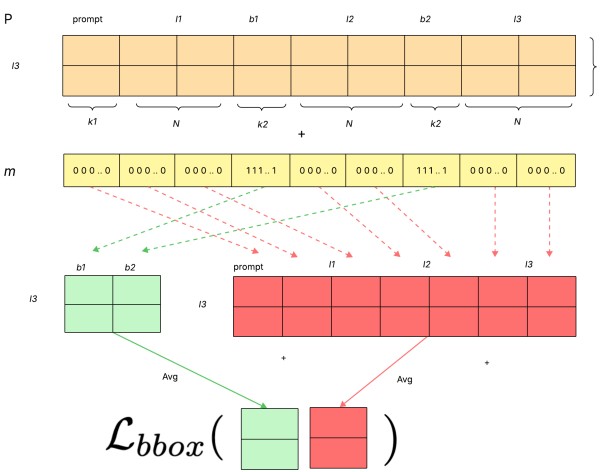

*Figure 4.* BBOX Attention Optimization: The mask for the BBOX token are given as 1 and remaining are 0 which is used to compute the average attention for the BBOX and non-BBOX token using the Eq-7.

instructed to predict the corresponding BBOX for the query image.

Formally, the complete input to the model is constructed as an interleaved sequence:

$$\mathcal{C} = \langle \texttt{prompt}, (I_1, b_1), \dots, (I_{T-1}, b_{T-1}), (I_T) \rangle \quad (4)$$

Note that Eq. 4 contains no category information; only visual inputs are provided to the model. The task is framed as in-context localization, where the target object is specified exclusively through BBOX demonstrations contained within the prompt.

### 5.2. Bounding Box Attention Optimization

In the LVM failure analysis (Section 3), we observe that existing models exhibit weak attention to the visual context of both support and query images, failing to sufficiently emphasize the regions critical for accurate localization. As a result, predictions are often dominated by prior semantic knowledge rather than instance-specific visual evidence, leading to incorrect localization and persistent category bias. To address this limitation, we explicitly optimize the model's latent attention maps to concentrate on the key spatial regions corresponding to the annotated support objects and the query image.

Let $x = (x_1, \dots, x_T)$ denote the input token sequence obtained from Eq. 4, consisting of the prompt and interleaved vision and text tokens from both the support set and the query example. The model $\mathbf{f}_\theta$ consists of a set of transformer layers. For each transformer layer $\ell \in \{1, \dots, L\}$ and attention head $h \in \{1, \dots, H\}$, the model produces an attention matrix $A^{(\ell,h)} \in \mathbb{R}^{T \times T}$. We aggregate attentions

across the layers and heads by averaging:

$$A = \frac{1}{LH} \sum_{\ell=1}^{L} \sum_{h=1}^{H} A^{(\ell,h)}. \quad (5)$$

Let $\mathcal{I}_{\text{query}} \subset \{1, \dots, T\}$ denote the index set of tokens corresponding to the query image. We extract the rows of $A$ indexed by $\mathcal{I}_{\text{query}}$, yielding

$$P \in \mathbb{R}^{N \times T} \quad (6)$$

where $N = |\mathcal{I}_{\text{query}}|$ is the number of query image tokens. Each row $P_i$ represents the attention distribution from query image token $i$ to all input tokens.

**Bounding Box Token Mask** During supervised fine-tuning, BBOX annotations for the support images are explicitly provided in the prompt as structured text (e.g., $[x_{min}, y_{min}, x_{max}, y_{max}]$). Since these annotations are serialized into text tokens by the tokenizer, the corresponding token indices are deterministically known. Based on this, we construct a binary mask $m \in \{0, 1\}^T$, where $m_j = 1$ if token $j$ corresponds to a BBOX annotation token from the support examples, and $m_j = 0$ otherwise. A schematic illustration of the token mask and the associated loss computation is shown in Fig. 4.

Equation 6 denotes the attention map from query image tokens to all input prompt tokens. We then compute the average attention mass assigned to BBOX tokens and to non-bounding box tokens:

$$p_i^+ = \frac{\sum_j P_{ij} m_j}{\sum_j m_j}, \qquad p_i^- = \frac{\sum_j P_{ij}(1 - m_j)}{\sum_j (1 - m_j)}. \quad (7)$$

The attention preference margin for query image token $i$ is defined as:
$$\Delta_i = p_i^+ - p_i^- \quad (8)$$

The loss for the margin $\Delta_i$ is formulated to encourage query image tokens to preferentially attend to BBOX annotation tokens by minimizing a margin-based hinge loss.

$$\mathcal{L}_{\text{bbox}} = \frac{1}{N} \sum_{i=1}^{N} \max(0, \mu - \Delta_i)^2, \quad (9)$$

where $\mu > 0$ is a hyperparameter. This loss enforces a relative preference for bounding box tokens over unrelated context, without constraining absolute attention magnitudes.

### 5.3. Supervised Fine-Tuning Objective

Let $\mathcal{L}_{\text{LM}}$ denote the standard language modeling loss. Incorporating the BBOX attention optimization the supervised fine-tuning objective is defined as:

$$\mathcal{L}_{\text{SFT}} = \mathcal{L}_{\text{LM}} + \beta \mathcal{L}_{\text{bbox}}, \quad (10)$$

where $\beta$, controls the strength of bounding box attention supervision which is optimized using the grid search over the validation data. This objective biases the model to ground its internal representations in spatial annotations provided by the support examples. Instead of training the model parameter $\theta$ we add the LoRA (Hu et al., 2021) weight $\phi$ and we only train the LoRA parameters.

While BBOX Attention Optimization emphasizes key regions in support and query images, it does not guarantee specific format and precise bounding box alignment. To address this limitation, we further optimize the model using Group Relative Policy Optimization (GRPO), encouraging accurate query bounding box prediction $b_{\text{pred}}$. The reward consists of two components: (i) an IoU-based reward computed against the query ground truth, and (ii) a formatting reward that enforces syntactic validity of the predicted bounding box.

## 5.4. Reinforcement Learning with GRPO

Group Relative Policy Optimization (GRPO) is a reinforcement learning method that updates policies using *relative comparisons* among multiple sampled responses, rather than a learned critic. In contrast to PPO (Schulman et al., 2017), GRPO (Shao et al., 2024; DeepSeek-AI, 2024) may relies on rule-based rewards and therefore avoids value function estimation. Given a query $q$, GRPO samples $G$ candidate outputs

$$\{o_1, o_2, \ldots, o_G\} \sim \pi_{\theta_{\text{old}}}(\cdot \mid q),$$

each of which is assigned a scalar reward, producing

$$\{r_1, r_2, \ldots, r_G\}.$$

The policy parameters $\theta$ are optimized by maximizing

$$
\begin{aligned}
\mathcal{J}_{\text{GRPO}}(\theta) =& \mathbb{E}\Bigg[\frac{1}{G}\sum_{i=1}^{G} \min\Bigg(\frac{\pi_\theta(o_i \mid q)}{\pi_{\theta_{\text{old}}}(o_i \mid q)} A_i, \\
& \text{clip}\Bigg(\frac{\pi_\theta(o_i \mid q)}{\pi_{\theta_{\text{old}}}(o_i \mid q)}, 1-\epsilon, 1+\epsilon\Bigg) A_i\Bigg)\Bigg] \\
& - \beta\, \mathcal{D}_{\text{KL}}(\pi_\theta \,\|\, \pi_{\text{ref}})
\end{aligned}
\tag{11}
$$

where $\epsilon$ controls the clipping range and $\beta$ weights KL regularization against a fixed reference policy $\pi_{\text{ref}}$. Advantages $A_i$ are computed *within each group* using normalized rewards:

$$A_i = \frac{r_i - \text{mean}(\{r_1, \ldots, r_G\})}{\text{std}(\{r_1, \ldots, r_G\})}.$$

This relative formulation promotes higher-quality responses within each sampled group and provides a stable alternative to critic-based optimization.

**Query IoU reward.** To ensure accurate localization with respect to the full query object, we additionally include the standard IoU between the predicted box and the query ground truth:

$$r_{\text{iou}} = \text{IoU}(b_{\text{pred}}, b_{\text{qry}}). \tag{12}$$

**Formatting reward.** Because bounding boxes are generated as text, we include a formatting reward $r_{\text{fmt}}$ that encourages syntactically valid predictions. A prediction is considered valid if it follows the required structure

$$\langle\texttt{answer}\rangle[x_{\min}, y_{\min}, x_{\max}, y_{\max}]\langle/\texttt{answer}\rangle.$$

The formatting reward is defined as

$$
r_{\text{fmt}}(\hat{y}) = \begin{cases} 0.5, & \text{if } \hat{y} \text{ matches the specified format,} \\ 0, & \text{otherwise.} \end{cases}
$$

**Combined reward.** The final reward ($\mathcal{R}$) used for GRPO is a weighted sum of the two above rewards:

$$\mathcal{R} = r_{\text{iou}} + r_{\text{fmt}}, \tag{13}$$

The reward $\mathcal{R}$ are optimized w.r.t. the LoRA parameters $\phi$. GRPO updates the policy by contrasting each trajectory's reward against a group-wise baseline computed over sampled trajectories, yielding low-variance gradients and stable optimization in the few-shot localization setting. GRPO assigns a higher advantage to predicted BBOXes that outperform the average reward, which strongly encourages the model to concentrate on the most accurate alignment during in-context few-shot training scenarios

## 6. Results and Discussions

The following section evaluates the proposed model across various datasets and compares its results with recent state-of-the-art baselines. Further, we investigate the proposed components and present the results in the ablations.

## 7. Dataset Details

We evaluate FOCUS for the in-context localization on a diverse collection of video and image benchmarks, including LaSOT (Fan et al., 2019), GOT-10k (Huang et al., 2019), TAO (Dave et al., 2020), PerSeg (Zhang et al., 2023), and PerMIRS (Samuel et al., 2024), which together span a wide range of object diversity, scene complexity, and generalization regimes. All datasets provide frame-level spatial annotations that we use to construct prompt-based in-context localization tasks, where the target object is specified implicitly via bounding-box demonstrations rather than semantic labels or explicit identity cues.

LaSOT is a large-scale object video benchmark comprising long sequences with dense, high-quality bounding-box annotations across 85 object categories. The extended temporal

*Table 1.* In-Context Few-shot localization performance (%) across multiple benchmark datasets and model variants.

| Model | TAO | | | GOT | | | ICL-LASOT | | | Avg. |
|---|---|---|---|---|---|---|---|---|---|---|
| | 1-shot | 2-shot | 4-shot | 1-shot | 2-shot | 4-shot | 1-shot | 2-shot | 4-shot | |
| Idefics3 | 6.8 | 15.0 | 25.5 | 9.3 | 21.2 | 32.3 | 3.6 | 8.7 | 14.7 | 15.2 |
| Pixtral-12B | 8.2 | 22.7 | 16.8 | 13.9 | 19.4 | 23.7 | 4.6 | 7.6 | 22.4 | 15.5 |
| LLaVA-OV | 22.5 | 29.5 | 33.5 | 18.6 | 26.4 | 33.7 | 10.8 | 14.1 | 17.7 | 23.0 |
| Qwen2-VL-7B | 26.0 | 31.6 | 36.1 | 36.2 | 37.0 | 39.3 | 26.2 | 22.3 | 25.0 | 31.1 |
| IPLoc (7B) | 51.7 | 54.3 | 56.1 | 64.2 | 68.1 | 68.7 | 49.7 | 57.1 | 59.4 | 58.8 |
| Qwen2-VL-72B | 46.2 | 52.8 | 55.6 | 62.7 | 60.1 | 59.9 | 51.9 | 50.7 | 55.4 | 55.0 |
| InternVL2-76B | 50.4 | 55.2 | 57.5 | 65.8 | 66.7 | 65.4 | 44.2 | 47.3 | 52.5 | 56.1 |
| FOCUS (Ours) | **55.8** | **63.0** | **68.5** | **77.1** | **80.6** | **82.6** | **56.1** | **59.9** | **65.6** | **67.7** |

duration of LaSOT videos introduces substantial appearance variation due to viewpoint changes, occlusions, and scale shifts, making it well-suited for evaluating localization robustness over time. GOT-10k emphasizes category-level generalization by enforcing a strict train–test split with zero overlap in object classes. As a result, models must localize objects from previously unseen categories at test time, which directly aligns with our in-context formulation, where object identity must be inferred solely from contextual visual and spatial cues. Both LaSOT and GOT-10k provide bounding box annotations for a single target object per sequence.

TAO represents a more challenging, open-world setting, with high-resolution videos containing multiple annotated objects per frame and a large, diverse vocabulary of object categories. Unlike LaSOT and GOT-10k, TAO includes multiple object instances within the same scene, often under significant clutter and occlusion. To construct a consistent in-context localization benchmark, we select the most frequently occurring object track within each video sequence as the target instance and treat the remaining objects as distractors. This setting evaluates a model's ability to resolve object identity from context alone in the presence of competing visual signals.

For all video datasets, we construct a fixed-shot in-context setting by uniformly sampling a set of support frames from each sequence. The first support frame is selected from the beginning of the video, the last from the end, and the remaining frames are sampled at equal temporal intervals in between. This strategy ensures that the in-context examples span the full temporal extent of the video and capture meaningful appearance variation of the target object. The final frame is used as the query, for which the model must predict the target object's bounding box.

To further assess generalization beyond natural video benchmarks, we also evaluate on PerSeg and PerMIRS, which provide segmentation annotations that we convert to bounding boxes. PerSeg is a synthetic dataset containing a single object per image and serves as a controlled setting for evaluating basic in-context localization behavior. In contrast,

*Table 2.* In context localization results on domain shift datasets

| Model | PerMIRS | | PerSeg | | | Avg. |
|---|---|---|---|---|---|---|
| | 1-shot | 2-shot | 1-shot | 2-shot | 4-shot | |
| Qwen2-VL-7B | 24.5 | 50.2 | 64.0 | 65.7 | 63.4 | 53.6 |
| IPLoc (7B) | 41.2 | 53.3 | 84.1 | 83.1 | 79.4 | 68.2 |
| Qwen2-VL-72B | 43.9 | 54.3 | 91.5 | 92.6 | 95.8 | 75.6 |
| FOCUS (Ours) | **53.8** | **72.8** | **95.5** | **96.6** | **97.5** | **83.2** |

PerMIRS contains scenes with multiple objects, often including several instances from the same category, increasing ambiguity and requiring finer-grained instance-level reasoning. Together, these datasets span a wide spectrum of visual complexity, ranging from PerSeg, the simplest scenario, to TAO, the most challenging setting, with an average of 4.1 objects per image.

Overall, this collection of datasets enables a comprehensive evaluation of in-context localization across controlled, category-generalization, and open-world scenarios.

### 7.1. Implementation Details

We fine-tune our models using LoRA; the details of the LoRA configuration are provided in the appendix. All experiments are run using the DeepSpeed distributed library for efficient distributed training. For our main results, we use Qwen2-VL-7B as the base model and fine-tune all models under a four-shot setting. We find that models fine-tuned under this setting generalize reasonably well to other shot configurations, and we therefore adopt the four-shot setting for all our experiments. We conducted all the inference (1-shot, 2-shot, 4-shot) using the same four-shot training model, demonstrating the model's generalization. We tuned the model's hyperparameters using a validation set. The Additional details about the model hyperparameters and other experimental details are provided in the appendix.

### 7.2. Attention Heatmap Upsampling and Visualization

To visualize attention distributions over the input image, we convert the model's 1D attention vector into a spatial

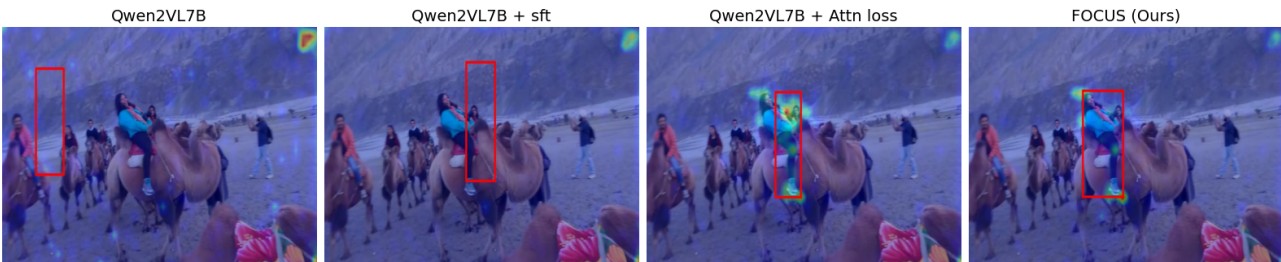

*Figure 5.* We share attention-based localization heatmaps across models and compare Qwen2-VL-7B under different training regimes. The vanilla model fails to localize the person riding the camel, while fine-tuning improves localization but remains incomplete. In contrast, our attention-based loss improves visual grounding and accurately localizes the target, with further gains from reinforcement learning.

heatmap aligned with the image's resolution. Given a 1D attention tensor of length $N$, we first infer its underlying 2D grid resolution $(H_{grid}, W_{grid})$ based on the target image dimensions $(H, W)$. The attention vector is then reshaped into a 2D map of size $H_{grid} \times W_{grid}$. To obtain a dense attention heatmap at the image resolution, we upsample this grid using bilinear interpolation to size $H \times W$, ensuring spatial alignment with the original image. This upsampled heatmap is subsequently normalized and overlaid on the image for visualization and analysis.

### 7.3. Evaluation Metrics

Similar to the IPLoc (Doveh et al., 2025) we use mIoU as the primary evaluation metric, reporting the IoU on the query image for the test set. We evaluate our model under different in-context few-shot settings. To demonstrate robustness across datasets, we fine-tune on multiple datasets and report performance on their respective test sets. In addition, we evaluate generalization by reporting performance on datasets outside the training distribution.

### 7.4. Results

We evaluate in-context localization performance across multiple datasets and compare our method against strong vision–language baselines, including Idefics3 (Laurençon et al., 2023), Pixtral-12B (Agrawal et al., 2024), LLaVA-OV (Li et al., 2024), Qwen2-VL-72B (Wang et al., 2024), InternVL2-76B (Chen et al., 2024), and IPLoc (). Results are reported on TAO (Dave et al., 2020), GOT (Huang et al., 2019), ICL-LaSOT (Fan et al., 2019), PerSeg (Zhang et al., 2023), and PerMIRS (Samuel et al., 2024) datasets. For each dataset, model hyperparameters are tuned using 10% of the training samples. On these datasets, the proposed model is evaluated under the following scenarios:

**In-Context Localization Generalization Capabilities:** To assess generalization beyond the training distribution, we evaluate FOCUS on two held-out datasets, PerMIRS and PerSeg. PerMIRS contains videos with multiple in-

stances from the same semantic category, while PerSeg is a synthetic dataset generated using a diffusion model. We report mean IoU (mIoU) across shot settings. The model is trained on the joint TAO, GOT, and ICL-LaSOT data under the four-shot setting and evaluated on PerMIRS and PerSeg. The results are reported in Table 2.

PerSeg is a synthetic dataset with relatively simple scenes containing a single object per sample. Nevertheless, IPLoc (Doveh et al., 2025) underperforms on this benchmark, as shown in Table 2. The performance gap widens further on PerMIRS, where each sample contains multiple instances from the same semantic category. In this setting, IPLoc struggles to correctly disambiguate the target instance, highlighting a limitation of category-conditioned localization in the presence of instance ambiguity. In contrast, FOCUS maintains strong performance by relying exclusively on visual context from in-context support images rather than semantic category names. As summarized in Table 2, FOCUS achieves absolute improvements of 19.1% on PerMIRS and 13.5% on PerSeg over IPLoc in the 2-shot setting.

**In Context Localization on Seen/Unseen Classes:** We have evaluated the model on the TAO dataset, and the results are shown in Table 1. TAO is highly challenging, contains the open word category, and each image contains multiple objects. For the multi-object scenario, where other models suffer, FOCUS outperforms recent baselines by a significant margin. Further, to evaluate generalization to unseen classes, we report results on the ICL-LaSOT and GOT datasets. LaSOT contains 70 object categories; we train on 35 categories and evaluate on the remaining 35 unseen categories. Similarly, GOT enforces a strict category-disjoint split between training and testing. Results are summarized in Table 1. We observe that FOCUS generalizes substantially better to unseen classes than IPLoc, achieving absolute gains of 13.9% on GOT and 6.2% on ICL-LaSOT in the four-shot setting. These results indicate that FOCUS learns category-agnostic visual grounding from in-context examples rather than relying on category-specific cues.

*Table 3.* Ablations over the various components of the TAO Dataset

| Model | 1-shot | 2-shot | 4-shot |
|---|---|---|---|
| Qwen2-VL-7B | 26.0 | 31.6 | 36.1 |
| Qwen2-VL-7B (sft) | 22.1 | 28.9 | 34.3 |
| Qwen2-VL-7B (grpo) | 19.4 | 26.4 | 32.1 |
| Qwen2-VL-7B (grpo+sft) | 24.3 | 27.2 | 33.7 |
| Qwen2-VL-7B (sft + attn loss) (ours) | 51.7 | 54.0 | 57.1 |
| Qwen2-VL-7B (sft + attn loss + grpo) (Ours) | **55.8** | **63.0** | **68.5** |

## 8. Ablations

We conduct extensive ablation studies to analyze the contributions of individual components in the proposed model. Specifically, we examine the attention distribution of answer tokens across different groups of input tokens, including query image tokens, support image tokens, and bounding box annotation tokens. We find the TAO dataset to be most suitable for these ablations, as it features open-vocabulary categories and multiple objects per frame. Compared to a naïve SFT baseline, our model assigns substantially higher attention to query image tokens while reducing reliance on bounding box annotation tokens. Although our training objective explicitly encourages query image tokens to attend to bounding box annotations, this behavior reflects representation learning induced by the attention loss rather than a failure of the objective. Quantitatively, as shown in Table 3, incorporating the attention loss alongside SFT yields significant performance gains. Further refining bounding box prediction using the GRPO loss improves boundary alignment, resulting in an additional 11.4% gain in the 4-shot setting over SFT with attention loss. Qualitative results in Fig. 5 further demonstrate that attention loss improves focus on relevant regions, while the combination of attention loss and GRPO enables accurate bounding box prediction in multi-object scenarios.

### FOCUS with a different base model

We evaluate the generality of our approach on an additional architecture, LLaVA-OV, using the TAO dataset. The results are shown in Table 4. FOCUS consistently improves over the LLaVA-OV baseline across all shot settings, increasing performance from 22.5 to 52.2 in the 1-shot setting, 29.5 to 59.7 in the 2-shot setting, and 33.5 to 65.7 in the 4-shot setting. These results show that our method is not limited to a single base VLM and can provide consistent gains across different architectures.

### Performance Across Different Layers of Attention

We ablate the effect of applying the attention loss at different layer groups: the first five layers, the last five layers, and the mean attention across all layers. We evaluate Qwen2 VL 7B on TAO in the 4-shot setting, as shown in Table 5. Applying

*Table 4.* Performance comparison with LLaVA-OV architecture on the TAO dataset: FOCUS shows consistently better results for the 1, 2, and 4-shot settings.

| Model | 1-shot | 2-shot | 4-shot |
|---|---|---|---|
| LLaVA-OV | 22.5 | 29.5 | 33.5 |
| FOCUS (on LLaVA-OV) | **52.2** | **59.7** | **65.7** |

*Table 5.* Ablation on attention loss placement across layers on the TAO dataset in the 4-shot setting.

| Model | First 5 | Last 5 | All |
|---|---|---|---|
| Qwen2-VL-7B (sft + attn loss) | **62.8** | 42.12 | 57.1 |

the attention loss to the first five layers achieves the best performance, with a score of 62.8, compared to 42.12 for the last five layers and 57.1 for all layers. This suggests that early-layer attention is more effective for grounding the model in the relevant visual region, while later layers primarily refine semantic features after the attended region has already been identified. This finding is interesting and could be a potential direction for future exploration. We conducted this study for completeness.

## 9. Memory Analysis with Attention Loss

During training, attention loss introduces a small memory overhead because attention maps must be retained for supervision. In our setting, SFT without attention loss requires 57.2 GB of VRAM, while adding the attention loss requires 64.1 GB. This corresponds to approximately 12% additional VRAM with a reasonable training batch size.

Importantly, this overhead is only incurred during training. At inference time, the attention loss is not used, so both memory usage and FLOPs remain the same as the base SFT model. Thus, our method improves grounding performance without adding any inference time computational cost.

## 10. Conclusions

In this work, we investigated in-context object localization under a category-agnostic, purely visual conditioning setting and identified key limitations of existing vision–language models, including reliance on semantic priors and weak spatial grounding. To address these issues, we proposed FOCUS, a two-stage framework that optimizes in-context attention over support bounding boxes and refines localization using Group Relative Policy Optimization. By enforcing reliance on visual correspondence rather than category supervision, our approach achieves robust instance-level localization. Extensive experiments demonstrate consistent improvements over strong baselines, including models an order of magnitude larger, highlighting the importance of context-aware localization objectives over model scale.

## Impact Statement

This work advances in-context visual understanding and may positively impact applications such as image editing, accessibility tools, and visual search. We expect its broader societal effects to align with established, beneficial uses of vision–language models, without introducing new ethical concerns.

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

## A. BBOX Notations

Following the standard convention adopted by recent vision–language models, bounding box coordinates are represented in a normalized coordinate space. Specifically, bounding boxes are scaled relative to the image dimensions and mapped to a fixed 0–1000 range. To remain consistent with this convention, we apply the same normalization to all bounding box annotations during preprocessing.

## B. Failure Cases:

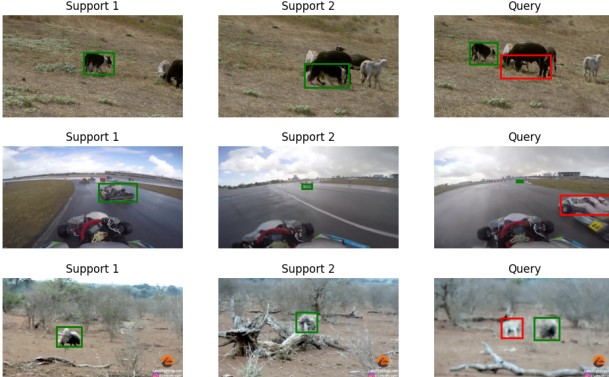

*Figure 6.* The figure shows representative 2-shot in-context localization failure cases on GOT. These examples illustrate the intrinsic difficulty of the task, where large viewpoint and scale changes between support and query images, heavy occlusion, and background clutter make reliable localization from a small number of support examples highly challenging.

## C. Hyperparameter Tuning for $\mu$ and $\beta$

During training with the attention loss, we perform a grid search over the margin parameter $\mu$ and the weighting factor $\beta$ on the TAO dataset under the 4-shot setting. As shown in Table 1, performance is sensitive to the choice of these hyperparameters, with intermediate values yielding substantially better localization accuracy. Based on this analysis, we select $\mu = 0.025$ and $\beta = 0.25$ for the TAO dataset. Similar hyperparameter searches are conducted independently for each dataset.

*Table 6.* Grid search over $\mu$ and $\beta$ on TAO in the 4-shot setting.

| $\mu \downarrow$ / $\beta \rightarrow$ | 0.0 | 0.01 | 0.15 | 0.20 | 0.25 | 0.30 | 0.35 | 1.0 |
|---|---|---|---|---|---|---|---|---|
| 0.025 | 34.3 | 37.2 | 53.2 | 55.9 | **57.1** | 56.4 | 52.5 | 28.1 |
| 0.05 | 34.3 | 40.8 | 48.6 | 49.7 | 50.2 | 51.3 | 50.7 | 24.2 |
| 0.075 | 34.3 | 38.7 | 46.1 | 47.3 | 47.5 | 45.5 | 44.7 | 23.4 |

## D. Attention Heatmap Upsampling and Visualization.

To visualize attention distributions over the input image, we convert the model's 1D attention vector into a spatial heatmap aligned with the image's resolution. Given a 1D attention tensor of length $N$, we first infer its underlying 2D grid resolution $(H_{\text{grid}}, W_{\text{grid}})$ based on the target image dimensions $(H, W)$. The attention vector is then reshaped into a 2D map of size $H_{\text{grid}} \times W_{\text{grid}}$. To obtain a dense attention heatmap at the image resolution, we upsample this grid using bilinear interpolation to size $H \times W$, ensuring spatial alignment with the original image. This upsampled heatmap is subsequently normalized and overlaid on the image for visualization and analysis.

*Table 7.* Training configuration for supervised fine-tuning (SFT), GRPO, and LoRA.

| Setting | Value |
|---|---|
| **Supervised Fine-Tuning (SFT) Configuration** | |
| Base model | Qwen2-VL-7B-Instruct |
| Learning rate | $2 \times 10^{-4}$ |
| Gradient accumulation steps | 16 |
| Warmup ratio | 0.03 |
| Max gradient norm | 0.3 |
| Precision | bfloat16 |
| **GRPO Training Configuration** | |
| Base model | Qwen2-VL-7B-Instruct |
| Optimization method | Group Relative Policy Optimization (GRPO) |
| Number of generations | 4 |
| Max prompt length | 8192 |
| Learning rate | $1 \times 10^{-5}$ |
| Warmup ratio | 0.01 |
| Max gradient norm | 1.0 |
| Precision | bfloat16 |
| Attention implementation | FlashAttention-2 |
| Reward functions | Accuracy, Format correctness |
| **LoRA Fine-Tuning Configuration** | |
| LoRA rank ($r$) | 8 |
| LoRA scaling ($\alpha$) | 16 |
| LoRA dropout | 0.0 |
| Target modules | `q_proj, k_proj, v_proj, o_proj, up_proj, down_proj, gate_proj` |
| Task type | Causal language modeling |
| Model quantization | 4-bit |

# E. FOCUS with 72B model

We further evaluate FOCUS on the 72B model setting to compare with the strongest reported IPLoc baseline. IPLoc reports results using Qwen2-VL-72B with real and pseudo examples. Since our original experiments were conducted under a more limited compute setting, we trained FOCUS primarily with Qwen2-VL-7B. To assess scalability to larger models, we additionally train FOCUS using Qwen3-VL-72B on 8 A100 80GB GPUs and evaluate on the ICL-LASOT dataset. The results are reported in Table 8.

*Table 8.* Comparison with IPLoc on TAO.

| Model | 2-shot | 4-shot |
|---|---|---|
| IPLoc (72B) (Real + Pseudo) | 65.71 | 67.63 |
| FOCUS (on Qwen2-VL-72B) | **70.72** | **72.35** |

