# OpenReview forum: "FOCUS: Forcing In-Context Object Localization through Visual Support Constraints and Policy Optimization"
_ICML.cc/2026/Conference — ICML 2026 regular_

### Official Review · Reviewer_wfDB · 2026-03-06

**Soundness:** 2
**Presentation:** 2
**Significance:** 2
**Originality:** 2
**Overall Recommendation:** 3
**Confidence:** 4

**Summary:**

This paper proposes a category-independent, pure visual context-based framework for in-context localization. It claims that current methods suffer from category-induced bias, which causes localization errors. To counter this, it proposes an attention-related loss that forces the query image to focus more intensely on bounding box annotations. It also leverages GRPO to fine-tune the model and reduce alignment discrepancies. Experimental results show that the proposed method leads to improved localization abilities.

**Compliance With Llm Reviewing Policy:**

Affirmed.

**Final Justification:**

The authors' rebuttal have addressed serveral of my questions. However, my main concerns remain:

First, I remain unconvinced by the novelty claim. The authors state that this is the first work to directly optimize attention in the latent space, but I find this claim too strong. In my view, the novelty here is limited and better characterized as a task-specific formulation.

Second, I am still unconvinced by the logic of Eq. 9. My concern is not whether bounding box coordinates can serve as annotations in general, but whether constraining attention toward annotation tokens, rather than the visual content inside the support regions, is fully aligned with the paper's stated motivation of enforcing visual grounding. The rebuttal does not fully resolve this conceptual gap. In particular, I do not think the analogy to standard object detection pipelines is sufficient to justify this design choice.

Third, I appreciate that this rebuttal introduces new experimental results. However, some of these results also raise further questions. For example, the newly reported layer-wise study suggests that supervising only the first 5 layers performs better than the all-layer setting used in the paper. This weakens the justification for the design choice adopted in the main submission, and may suggest that the ablation in the original paper was not yet sufficiently mature.

Considering the above issues, I keep my original rating.

**Key Questions For Authors:**

See the Weaknesses.

My initial recommendation is Weak Reject. I will consider raising my recommendation score if the authors can address my concerns convincingly.

**Limitations:**

There is a Limitation section in the suppl.

**Strengths And Weaknesses:**

**Strengths:**

1. The proposed model achieves significant improvement.
2. The loss design is simple and easy to implement.


**Weaknesses:**

1. Limited Novelty: The model design and training data are standard. The main contribution is limited to the attention constraint in Eq. 9.

2. Logical Flaw in Eq. 9: The equation constrains the attention between the query image and the bounding box annotations. However, should it reference the specific regions within the support images defined by those boxes, not the annotations themselves, as we want the model to read the **visual content** to locate the target in the query image?

3. In the original paper, IPLoc-72B reports a performance of 67.63. It is not clear to me why this paper only compares with IPLoc-7B in Table 1. As a result, the scalability of the proposed method remains uncertain.

4. The ablation studies are limited and leave several critical questions unanswered:
    a) Hyper-parameter Sensitivity: How is the important hyper-parameter `\mu` determined? Is it tuned via grid search, or is there a theoretical basis for its selection?
    b) Performance Degradation: Why do both the SFT-only and SFT+GRPO variants underperform compared to the baseline? An analysis of this performance drop is necessary to understand the contribution of each component.
    c) Layer Aggregation Strategy: Eq. 5 employs simple uniform averaging across all layers. This lacks in-depth analysis. Specifically, the paper should investigate how using different subsets of layers (e.g., shallow vs. deep layers) affects the performance, rather than assuming uniform contribution."


**Other Problems:**

1. Inconsistent Presentation: There seems to be a conceptual gap between the Method section (which describes the attention flow from Query→Box) and Figure 2 (which visualizes the attention from Answer→Input). The authors should clarify the relationship between the designed mechanism and the visualized patterns to ensure logical consistency.

2. Does Eq. 3 miss the system prompt?

3. Please resolve the notation conflict for the symbol `T`. It is currently used to represent both the number of shots (Eq. 4) and the number of tokens (Eq. 6). Distinct symbols should be assigned to different variables.

---

> ### Author Rebuttal · Authors · 2026-03-30
>
> We thank the reviewer for the time and constructive feedback. Please find the response to each point in detail.
>
> **W1R1:** We politely disagree with the reviewer regarding the novelty aspect. While Eq. 9 defines the final optimization objective, to the best of our knowledge, the complete formulation in Sec. 5.2 is novel. In particular, the formulations in Eq. 7 and Eq. 8, where we define the positive and negative attention along with the margin, are also highly important. To the best of our knowledge, this is the first work that directly optimizes attention in the latent space instead of focusing only on label-based objective optimization. We request the reviewer to kindly point us to similar prior work so that we can provide a more concrete response. Also, applying reward-based optimization is not straightforward; we use a Query IoU-based reward and optimize it using GRPO.
>
> **W2R2:** We disagree with the reviewer that this is a logical flaw. This is a common practice in the object detection literature, where a region is represented by the bounding box coordinates $<x_1, y_1, x_2, y_2>$ (e.g., FasterRCNN, YOLO, DETR), including in the baseline IPLoc paper itself. The same coordinates can be used to constrain the attention map and optimize the objective function. Also, in a transformer, since each token interacts with every other token, this optimization helps the model emphasize the key visual tokens while ignoring the remaining ones.
>
> **W3R3:** Thanks for pointing this out. The baseline paper reports results for IPLoc-72B, which uses the Qwen2-VL-72B model. Because we had limited resources (8xV100, 32GB), this GPU setup was not sufficient to support 72B parameters. Therefore, for fair comparison, we trained the proposed model only on the Qwen2-VL-7B setting and compared against the corresponding baseline. As suggested by the reviewer, we also conducted experiments on the Qwen2-VL-72B architecture (using 8xA100, 80GB GPUs) as a baseline on the ICL-LASOT dataset and report the results below:
>
> | Model | 2-shot | 4-shot |
> |---|---:|---:|
> | IPLoc (72B) (Real+Pseudo) | 65.71 | 67.63 |
> | FOCUS (on Qwen3-VL-72B) | 70.72 | 72.35 |
>
>
> We will add the detailed version in the updated paper. We also observe that scalability is not a significant issue, since during inference the model requires the same memory and FLOPs as the original model.
>
> **W4R4:**
>
> ***a-*** The hyperparameters are tuned using grid search over the validation data for each dataset independently. The results over $\mu$ and $\beta$ on the TAO dataset are reported below:
>
> Grid search over $\mu$ and $\beta$ on TAO (4-shot) for Qwen2-VL-7B (sft + attn loss):
>
> | $\mu \downarrow$ / $\beta \rightarrow$ | 0.00 | 0.01 | 0.15 | 0.20 | 0.25 | 0.30 | 0.35 | 1.0 |
> |---|---:|---:|---:|---:|---:|---:|---:|---:|
> | 0.025 | 34.3 | 37.2 | 53.2 | 55.9 | 57.1 | 56.4 | 52.5 | 28.1 |
> | 0.05  | 34.3 | 40.8 | 48.6 | 49.7 | 50.2 | 51.3 | 50.7 | 24.2 |
> | 0.075 | 34.3 | 38.7 | 46.1 | 47.3 | 47.5 | 45.5 | 44.7 | 23.4 |
>
> ***b-*** We thank the reviewer for pointing out this important observation. A similar question was also raised by reviewer "Md8F", where we provided a detailed answer based on our analysis. We kindly request the reviewer to please refer to the rebuttal "W1R1" in our response to reviewer "Md8F".
>
> ***c-*** We appreciate the reviewer for this interesting observation. We conducted an experiment to evaluate the same, and the results are discussed in our response "W3R3" to reviewer "Md8F" We kindly request the reviewer to please refer to that response.
>
>
> **W5R5: Inconsistent Presentation**
> The apparent inconsistency arises because the Method and Figure 2 examine two complementary but distinct views of the model. In the Method (Sec. 5.2), our objective explicitly enforces that query image tokens preferentially attend to BBOX annotation tokens (i.e., Query $\rightarrow$ BBOX), encouraging the model to ground its predictions in the visual support annotations. In contrast, Figure~2 presents a diagnostic visualization of attention from the answer token to the input tokens (Answer $\rightarrow$ Input), which reflects how the final prediction aggregates information across the context. These two views are not contradictory: the Answer $\rightarrow$ Input attention is a downstream consequence of the representations shaped by the Query $\rightarrow$ BBOX objective. As a result, the improved grounding enforced during training manifests as increased reliance on the relevant visual context in the answer attention maps.
>
> **W6R6:** Thanks for pointing out this mistake. We missed the prompt in Eq. 3, and we will update it accordingly. We will also resolve the notation conflict involving $T$.
>
> Please let us know if the reviewer requires any further clarification.

---

> > ### Author Rebuttal · Reviewer_wfDB · 2026-04-05
> >
> > I thank the authors for the detailed rebuttal and the additional experiments. I appreciate the effort taken during the rebuttal period.
> >
> > While the rebuttal has addressed several of my questions, my main concerns remain.
> >
> > First, I remain unconvinced by the novelty claim. The authors state that this is the first work to directly optimize attention in the latent space, but I find this claim too strong. Recent works have already explored improving grounding by supervising or constraining internal explanations, attention, or alignment, for example:
> > - Visual Grounding with Attention-Driven Constraint Balancing, ACM MM 2024.
> > - Improving Visual Grounding by Encouraging Consistent Gradient-Based Explanations, CVPR 2023.
> > - Improved Visual Grounding through Self-Consistent Explanations, CVPR 2024.
> > - Your Large Vision-Language Model Only Needs A Few Attention Heads For Visual Grounding, CVPR 2025.
> > - Direct Visual Grounding by Directing Attention of Visual Tokens, WACV 2026.
> >
> > While I am not stating that the proposed formulation is identical to these existing works, they make it difficult to regard the broader idea of directly supervising attention for grounding as novel. In my view, the novelty here is limited and better characterized as a task-specific formulation.
> >
> > Second, I am still unconvinced by the logic of Eq. 9. My concern is not whether bounding box coordinates can serve as annotations in general, but whether constraining attention toward annotation tokens, rather than the visual content inside the support regions, is fully aligned with the paper's stated motivation of enforcing visual grounding. The rebuttal does not fully resolve this conceptual gap. In particular, I do not think the analogy to standard object detection pipelines is sufficient to justify this design choice.
> >
> > Third, I appreciate that this rebuttal introduces new experimental results. However, some of these results also raise further questions. For example, the newly reported layer-wise study suggests that supervising only the first 5 layers performs better than the all-layer setting used in the paper. This weakens the justification for the design choice adopted in the main submission, and may suggest that the ablation in the original paper was not yet sufficiently mature.
> >
> > Overall, although I appreciate this rebuttal's effort and acknowledge that the method appears practically useful, the paper does not seem to be ready for publication yet. As a result, I keep my original recommendation.

---

> > > ### Author Response · Authors · 2026-04-06
> > >
> > > Thank you for the detailed feedback. We sincerely appreciate the reviewer’s time, insights, and thoughtful evaluation of our work. We respect the reviewer’s perspective and would like to offer a few clarifications that we hope help address the remaining concerns.
> > >
> > > We agree that prior work has explored the use of attention as a constraint. However, our formulation differs in a fundamental way: we explicitly define a margin and distinguish between bounding box (bbox) tokens and non bounding box tokens. In addition, existing approaches do not directly extend to the in context learning setting we consider. Finally, we emphasize that our GRPO formulation, defined over IoU, is a key contribution of this work and demonstrates clear improvements over standard attention based constraints.
> > >
> > > Our paper proposes a simple and intuitive approach to guiding attention from bounding boxes to visual tokens, inspired by object detection literature. While we acknowledge that there are multiple valid ways to achieve this objective, the existence of alternative methods does not diminish the validity of our approach. Both qualitative and quantitative results (see Figures 2 and 5) consistently show that our formulation effectively increases attention over relevant visual tokens. The motivation presented in the paper is aligned with these empirical findings.
> > >
> > > Regarding the design choice of aggregating attention across all layers, this was a deliberate decision supported by our experimental results, which indicate improved performance. We appreciate the reviewer’s suggestion and agree that other design alternatives may further enhance results. However, this is a general characteristic of most research works, there is often room for improvement through alternative design choices. We believe this should not be viewed as a weakness or a sign of immaturity, but rather as an indication of promising directions for future work.
> > >
> > > We hope this clarification helps address the reviewer’s concerns.

---

### Official Review · Reviewer_Md8F · 2026-03-09

**Soundness:** 2
**Presentation:** 3
**Significance:** 3
**Originality:** 2
**Overall Recommendation:** 4
**Confidence:** 4

**Summary:**

The paper proposes FOCUS, a two-stage training framework for In-Context Object Localization (ICOL) that eliminates the reliance on explicit category names in prompts. The authors identify that existing Vision-Language Models (VLMs) often exploit semantic priors (category labels) rather than genuine visual correspondence, leading to failure when multiple instances of the same category are present. To force true visual grounding, FOCUS first employs an attention-based margin loss that encourages query image tokens to explicitly attend to support bounding box tokens. In the second stage, it refines the localization policy using GRPO, utilizing formatting and IoU rewards without needing a critic model.

**Compliance With Llm Reviewing Policy:**

Affirmed.

**Final Justification:**

Given the responses from authors and the methodological novelty, I will maintain my current score at weak accept.

**Key Questions For Authors:**

1. See Weaknesses.
2. Can authors provide empirical results applying the FOCUS pipeline (Attention Loss + GRPO) to at least one other distinct VLM architecture to prove this is a generalized framework rather than a model-specific trick?

**Limitations:**

yes

**Strengths And Weaknesses:**

Pros:
1. The motivation to decouple visual grounding from semantic priors is valid, and the empirical observation that category tokens hijack attention mechanisms (Figure 2) is a well-founded starting point. This method has practical significance for application.

Cons:
1. The ablation study (Table 3) exposes a severe instability: applying GRPO alone to the base model degrades performance below the baseline (e.g., dropping from 26.0% to 19.4% in 1-shot). The authors fail to provide any rigorous theoretical or empirical diagnosis of this policy collapse.
2. The paper completely omits an analysis of the computational and memory overhead introduced by the attention loss. Extracting, storing, and computing gradients for the full attention graph imposes a massive VRAM burden during training, especially with long multimodal context windows. The feasibility of this approach in practical computing environments is not discussed.
3. The authors simply average the attention matrices across all layers and heads evenly. This naive pooling ignores the well-established layer-wise hierarchy in Transformers, where shallow layers capture low-level, local features, and deeper layers resolve high-level semantics. Forcing early layers and non-semantic attention heads to align with high-level BBOX annotations may disrupt the base model's foundational representation capabilities.

---

> ### Author Rebuttal · Authors · 2026-03-30
>
> We thank the reviewer for the time and constructive feedback. Please find the response to each point in detail.
>
> **W1R1:** We thank the reviewer for raising this question; we had a similar concern during our analysis. A similar pattern is also observed in the baseline paper IPLoc. Please refer to the Table-2 Qwen2-VL-7B and IPLoc (7B) (Real, i.e., SFT) results on the ICL-LASOT dataset, where the authors also observe that SFT degrades model performance. Although this is counter-intuitive, we performed a deeper analysis and have the following observations:
>
> **1.** We observe that, in fine-grained multi-object settings and long-tail distributions (e.g., TAO (Table-3), which contains long-tail classes (500 samples for training and 2400 for evaluation), and where the number of test samples from the tail is larger than that in the training data (TAO paper [a], Table-1)), SFT fine-tuning often fails. However, the base model is trained on trillions of tokens and therefore shows better generalization over long-tail classes. GRPO also optimizes an objective similar to SFT (Eq. 12); the only difference is that its optimization is advantage-based, and thus it behaves similarly to the SFT objective on long-tail datasets.
>
>
> **2.** Zero-shot objects (e.g., ICL-LASOT) represent another scenario where SFT and similar objectives fail. The SFT or GRPO model is not able to generalize to novel classes, and therefore shows lower results compared to the baseline model. The baseline Qwen2-VL-7B is a strong pretrained model trained on trillions of tokens, so it shows high generalization ability since most of the classes are not truly novel for this model. However, once we fine-tune the model for the in-context data, the model loses its generalization ability and is not able to predict unseen classes, or gets confused among similar classes.
>
> In our analysis, we observe that the above two points are the key reasons for the failure of the SFT-only and GRPO-only models. However, attention-based optimization ignores class information and focuses only on the visual context, and is therefore able to handle the above challenging scenarios. We will add this detailed analysis in the updated version.
>
> **W2R2:**
>
> ***Memory Analysis:***
>
> GPU: 8xV100 32GB
>
> Batch-Size: 64
>
> Bit Precision: 16 Bit
>
> Without attention (SFT): 57.2 GB VRAM
>
> With attention: 64.1 GB VRAM
>
> We observe that, with a reasonable batch size during training, our approach requires 12\% more VRAM. However, during inference, memory usage is the same in both settings since we do not need to compute the attention. Also, during inference, the FLOPs are the same for both settings.
>
> **W3R3:**
>
> Thanks for the suggestion. We conducted the suggested experiment and applied the attention loss only on the initial five layers and the last five layers, and compared it to the mean attention over all layers. The model is evaluated on the TAO dataset in the 4-shot setting, and the results are given in the table below:
>
> | Model | First 5 layers | Last 5 layers | All layers |
> |---|---:|---:|---:|
> | Qwen-2-VL-7B (SFT + attn loss) | 62.8 | 42.12 | 57.1 |
>
> We observe that applying the attention loss only on the initial five layers gives significantly better results compared to the last five layers or all layers. This suggests that early attention is more important than later attention, similar to how early fusion allows more effective deeper interactions. Once the model identifies the attention region in later stages, it mostly focuses only on the semantic features within that region. We will add this detailed analysis in the updated version.
>
> **W4R4: Result on LLaVA-OV**
> We conducted experiments on another architecture (LLaVA-OV) and report the results on the TAO dataset. Also, in our response to reviewer "YpdH", we have shown results on the Qwen3-VL-8B-Instruct'' architecture; please refer to that response as well. The results using "LLaVA-OV" as the base model are shown below:
>
> | Model | 1-shot | 2-shot | 4-shot |
> |---|---:|---:|---:|
> | LLaVA-OV | 22.5 | 29.5 | 33.5 |
> | FOCUS (on LLaVA-OV) | 52.2 | 59.7 | 65.7 |
>
> We observe that the model consistently shows improvement over the base model, following a pattern similar to what we observe for the Qwen architecture.
>
> [a] TAO: A Large-Scale Benchmark for Tracking Any Object, ECCV-2020
>
> Please let us know if the reviewer requires any further clarification.

---

> > ### Author Rebuttal · Reviewer_Md8F · 2026-04-02
> >
> > I thank the authors for their detailed rebuttal and the additional experiments. The response provides helpful clarifications and addresses the specific technical questions raised in my initial review:
> >
> > The new experiment confirming that the attention margin loss is primarily effective in the early layers rather than all layers resolves the structural concern regarding Transformer hierarchies. The additional results on LLaVA-OV provide acceptable evidence that the framework is not strictly limited to the Qwen architecture. The clarification that the attention extraction introduces a 12% VRAM overhead during training, with no inference cost, adequately addresses the memory bottleneck concern.
> >
> > Given the methodological novelty and overall significance, I will maintain my current score.

---

### Official Review · Reviewer_YpdH · 2026-03-10

**Soundness:** 2
**Presentation:** 3
**Significance:** 2
**Originality:** 2
**Overall Recommendation:** 4
**Confidence:** 2

**Summary:**

This paper introduces a two-stage training framework that explicitly optimizes in-context attention between support bounding boxes and query images without category supervision. Comprehensive ablations validate the contribution of each component.

**Compliance With Llm Reviewing Policy:**

Affirmed.

**Final Justification:**

The concerns I raised have been addressed by authors, so I would like to raise my socre to weakly accept.

**Key Questions For Authors:**

Why the author only compare Qwen2? It seems like a little bit out of date.

**Limitations:**

Please refers to "Weakness" and "Key Questions For Authors" parts.

**Strengths And Weaknesses:**

Stength:

1.The whole paper is well-written and easy to follow.

2. The ablation study is in detailed, and I can see the effectiveness of each module straightforward.

3. The visualisation cases are clear and easy to catch the idea of the paper.

Weakness:

1. The motivation for incorporating an RL-based training stage lacks robust justification. It remains conceptually unclear whether the observed performance gains are uniquely attributable to the RL framework or if comparable improvements could be achieved through more straightforward supervised learning objectives. The authors are encouraged to provide a comparative ablation study or a theoretical discussion to clarify the necessity of the RL component over simpler alternatives.

2. The evaluation is restricted to a specific ICOL setting, and it is unclear whether the proposed approach generalises well to broader grounding or instance retrieval tasks.

3. The manuscript posits that prior semantic knowledge may introduce undesirable biases that hinder the model’s focus on support examples. However, such priors often serve as valuable inductive biases for complex visual reasoning rather than inherent detriments. The paper would benefit from a more rigorous analysis demonstrating why these priors are specifically counterproductive within the ICOL context. Specifically, the authors should provide empirical evidence or qualitative examples where semantic priors lead to erroneous localizations despite the availability of clear, relevant support information.

---

> ### Author Rebuttal · Authors · 2026-03-30
>
> We thank the reviewer for the time and constructive feedback. Please find our response to each point below. As suggested by the reviewer, we also conducted experiments on the QWEN3-8B architecture and report the corresponding results.
>
> **W1R1:** We would like to draw the reviewer’s attention to Table-3 for the quantitative results and Figure-5 for the qualitative results. Table-3 reports the results for supervised fine-tuning (SFT), GRPO-based optimization, and attention-based optimization. We observe that SFT or GRPO alone degrades model performance; however, combining attention loss with SFT or SFT+GRPO significantly improves performance (the reason for this behavior is discussed in our response to reviewer "Md8F" under "W1R1"). The qualitative results in Figure-5 further show that FOCUS helps identify robust attention and capture correct, tight boundaries.
>
> **W2R2:** We agree with the reviewer that the paper explores only the IOCL setting; however, within this setting, we study several challenging scenarios:
>
> **1.** ICL-LASOT and GOT datasets contain a single object per frame but have disjoint train and test classes (zero-shot). The strong performance of the proposed model demonstrates its generalization ability beyond the training classes.
>
> **2.** We also report results on PerMIRS and TAO (which contain long-tail class distributions, and where most test samples lie in the tail region [a]). In these datasets, each frame contains multiple objects from the same category, i.e., they are fine-grained datasets that are highly challenging, with a high chance of confusion among similar objects.
>
> We believe that these settings are more important for evaluating the model’s generalization ability. If a model is able to detect and localize the object, the retrieval task becomes straightforward. We are happy to include retrieval results in the updated version.
>
>
> **W3R3:** We agree with the reviewer that, in LLMs or LVMs, "priors often serve as valuable inductive biases for complex visual reasoning rather than inherent detriments". However, in in-context learning, where the task involves fine-grained or zero-shot objects, the same prior can significantly degrade model performance. For example, in Figure-1, the support image contains a light-blue bowl, but if the model relies on the prior of "bowl", it may localize a white bowl instead of the given blue bowl. The prior may be helpful when only a single object is present, but even in such cases, the model will not work well in the zero-shot setting where train and test classes are disjoint (ICL-LASOT and GOT datasets), since novel classes do not have any prior. Therefore, the claim that "prior helps" is mostly true for single-object settings with overlapping train and test classes, but it does not generalize to zero-shot or fine-grained settings.
>
> **Comparison with the Qwen3-VL-8B-Instruct:** Thanks for the suggestion. As suggested by the reviewer, we consider Qwen3-VL-8B-Instruct as a baseline and evaluate the model on the TAO dataset under the IOCL setting. The results are given as follows:
>
> | Model | 1-shot | 2-shot | 4-shot |
> |---|---:|---:|---:|
> | Qwen3-VL-8B-Instruct | 27.6 | 33.8 | 39.3 |
> | FOCUS (on Qwen3-VL-8B) | 58.3 | 65.7 | 69.7 |
>
> [a] TAO: A Large-Scale Benchmark for Tracking Any Object, ECCV-2020
>
> Please let me know if reviewer have any further clarification.

---

> > ### Author Rebuttal · Reviewer_YpdH · 2026-04-03
> >
> > My concerns are addressed well, I will raise my score.

---

### Decision · Program_Chairs · 2026-04-30

**Decision:**

Accept (regular)

**Comment:**

Summary of reviews:
* YpdH (WA): (S1) The paper is well written; (S2) detailed ablation; (S3) clear visualization. (W1) The motivation for RL-based training lacks justification; (W2) evaluation is restricted to a specific ICOL setting; (W3) request for more evidence that semantic knowledge leads to undesirable biases. Concerns addressed by rebuttal.
* Md8F (WA): (S1) Solid motivation to decouple visual grounding from semantic priors. (W1) applying GRPO to base model drops performance (this isn't surprising to the AC, as SFT is typically required first); (W2) omits analysis of computational/memory cost of attention loss; (W3) simple pooling of attention across heads. Satisfied with author response.
* wfDB (WR): (S1) Good improvement; (S2) elegant loss. (W1) Main original contribution limited to attention constraint; (W2) Flaw in Eq 9; (W3) no comparison with larger model; (W4) missing ablations. Considered the author's response, but notes several remaining important issues.

Considering the weaknesses raised by wfDB: (1) AC does not believe there is a novelty problem. The paper offers a simple way to encourage the model to attend to bounding box annotations. Simple innovations that are very effective are preferred over complex ones. (2) AC sees the point about Eq 9 -- it is more intuitive to force attention to the visual tokens in the bbox rather than the bounding box tokens, and it would have been nice to see an ablation comparing them. However, AC does not see this as a critical flaw. (3) AC considers W3 and W4 to be adequately addressed by the responses.

Considering the strengths and weaknesses overall, the AC recommends to accept.

-------
Note that two reference issues were flagged.  At least one seemed to correspond to a similarly-titled paper, so AC did not report as AI hallucinations.  Please correct.
Reference: Zhou, X., Zhang, Y., Zhang, S., and Zhang, S. One-shot semantic segmentation. In British Machine Vision Conference (BMVC), 2018. URL https://arxiv.org/abs/1709.03410.
Issue: authors+title mismatch with arXiv

Reference: Wang, X., Huang, T. E., Gonzalez, J. E., Darrell, T., and Yu, F. Meta-learning for few-shot object detection. In Proceedings of the IEEE/CVF International Conference on Computer Vision (ICCV), 2019. URL https://arxiv.org/abs/1909.13032.
Issue: authors+title mismatch with arXiv